# Iron Accumulation and Changes in Cellular Organelles in WDR45 Mutant Fibroblasts

**DOI:** 10.3390/ijms222111650

**Published:** 2021-10-28

**Authors:** Hye Eun Lee, Min Kyo Jung, Seul Gi Noh, Hye Bin Choi, Se hyun Chae, Jae Hyeok Lee, Ji Young Mun

**Affiliations:** 1Neural Circuit Research Group, Korea Brain Research Institute, Daegu 41068, Korea; heni91@kbri.re.kr (H.E.L.); j0312@kbri.re.kr (M.K.J.); nsguu8676@kbri.re.kr (S.G.N.); 2Department of Brain and Cognitive Sciences, Daegu Gyeongbuk Institute of Science and Technology, Daegu 42988, Korea; 3Neurovascular Unit Research Group, Korea Brain Research Institute, Daegu 41068, Korea; hbchoi@kbri.re.kr; 4Department of Neurology, Research Institute for Convergence of Biomedical Science and Technology, Pusan National University Yangsan Hospital, Yangsan 50612, Korea; jhlee.neuro@pusan.ac.kr; 5Medical Research Institute, Pusan National University School of Medicine, Yangsan 50612, Korea

**Keywords:** iron overload, WDR45, lipid metabolism, mitochondria, lysosome, autophagy, CLEM

## Abstract

Iron overload in the brain, defined as excess stores of iron, is known to be associated with neurological disorders. In neurodegeneration accompanied by brain iron accumulation, we reported a specific point mutation, c.974-1G>A in WD Repeat Domain 45 (*WDR45*), showing iron accumulation in the brain, and autophagy defects in the fibroblasts. In this study, we investigated whether fibroblasts with mutated WDR45 accumulated iron, and other effects on cellular organelles. We first identified the main location of iron accumulation in the mutant fibroblasts and then investigated the effects of this accumulation on cellular organelles, including lipid droplets, mitochondria and lysosomes. Ultrastructure analysis using transmission electron microscopy (TEM) and confocal microscopy showed structural changes in the organelles. Increased numbers of lipid droplets, fragmented mitochondria and increased numbers of lysosomal vesicles with functional disorder due to *WDR45* deficiency were observed. Based on correlative light and electron microscopy (CLEM) findings, most of the iron accumulation was noted in the lysosomal vesicles. These changes were associated with defects in autophagy and defective protein and organelle turnover. Gene expression profiling analysis also showed remarkable changes in lipid metabolism, mitochondrial function, and autophagy-related genes. These data suggested that functional and structural changes resulted in impaired lipid metabolism, mitochondrial disorder, and unbalanced autophagy fluxes, caused by iron overload.

## 1. Introduction

Iron significantly contributes to brain homeostasis. Although iron is essential for proteins to function normally, iron overload can not only negatively impact brain development, but also induce brain injury. The concentration of iron should be tightly regulated, so that an appropriate amount of iron ions can be supplied for various cellular functions. The iron plays a role in mitochondrial respiration and the synthesis of important components such as myelin, neurotransmitters, and monoamine oxidases in the brain. However, studies on the relationship between iron imbalance and the function of cellular organelles, including the synthesis and turnover of cellular components, have been few. Neurodegeneration with brain iron accumulation (NBIA) is a group of inherited neurological disorders characterized by abnormal iron accumulation in the basal ganglia [1,2]. Beta-propeller protein-associated neurodegeneration (BPAN) caused by mutations in the autophagy gene WD Repeat Domain 45 (*WDR45)*, which encodes WD-repeat protein interacting with phosphoinositides 4 (*WIPI4*) has emerged as the most common NBIA disorder [2,3]. It is characterized by childhood developmental delay and seizures, followed by parkinsonism and dystonia, as well as dementia, in adolescence or early adulthood [2,3,4]. The *WDR45* mutation has been linked to defective autophagic flux in the cells of patients [5,6]. Consistent with these findings, reduced *WDR45* levels are correlated with increased iron levels, oxidative stress and mitochondrial dysfunction, as observed mainly via biochemical analysis in patient-specific *WDR45* mutant fibroblasts and induced pluripotent stem cell-derived midbrain neurons [7]. Defective neuronal autophagy and neurological abnormalities in *WDR45* knockout mice were reported [5,6].

The phenotypes that result from mutations in the *WDR45* gene indicate the importance of these proteins for neuronal functions. However, the association between abnormalities in autophagy and mitochondrial dysfunction related to iron accumulation is not fully understood. Therefore, we investigated iron accumulation in *WDR45* mutant fibroblast cells from a patient and studied the relationship between iron accumulation, defects in autophagy and mitochondrial dysfunction through structural and functional analysis. In our previous paper, we reported iron overload in the brain, identified using Magnetic Resonance Imaging (MRI) imaging and defects of autophagy in the fibroblasts of a patient. The patient had a heterozygous, splice-acceptor site mutation in the *WDR45* gene (c.977-1G>A; NM_007075.3, c.974-1G>A; NM_001029896.2) [8], which has been reported to be pathogenic (PVS1 PM2 PM6 PP3 PP5 according to the American College of Medical Genetics and Genomics Guidelines), listed in ClinVar database ((https://www.ncbi.nlm.nih.gov/clinvar/) (rs1557083830). This sequence change affects an acceptor splice site in the last intron (intron 11) of the *WDR45* gene. In this study, iron accumulation was found to be the most in specific lysosomes of *WDR45* mutant fibroblasts from the patient, and slight iron accumulation was observed in the mitochondria possessing the c.974-1G>A mutation. Iron accumulation can lead to ROS production, the generation of oxidative stress, dysfunction of the mitochondria, and abnormal autophagy. To investigate the structural and functional changes in autophagy and mitochondria caused by iron accumulation, such changes were measured using confocal microscopy, electron microscopy, and gene expression profiling analysis.

## 2. Results

### 2.1. Iron Accumulation Was Confirmed in WDR45 Mutant Fibroblasts

In our previous study, excessive brain iron accumulation in the patient with c.974-1G>A mutation was confirmed in vivo by magnetic resonance imaging [8]. We then examined iron accumulation and localization in *WDR45* mutant fibroblasts. First of all, we measured the abundance of WDR45 in mutant and normal fibroblasts (Figure 1a) using Western blotting to confirm WDR45 deficiency in *WDR45* mutated fibroblasts from the patient. The level of WDR45 was significantly reduced in *WDR45* mutant fibroblasts compared with levels in normal fibroblasts (Figure 1a).

To determine iron accumulation in cells, we measured the puncta of a fluorescent Fe^2+^ probe in *WDR45* mutant fibroblasts from the patient using confocal microscopy. Compared with normal fibroblasts, the mutant cells showed numerous puncta, and it was a significant 2.6-fold increase in the fluorescence intensity of the Fe^2+^ probe (Figure 1b). To study the location of the puncta, we compared the puncta to lipid and lysotracker signals (Figure 1c). As shown in Figure 1c, the major Fe^2+^ signal (red) was compatible with lysotracker (blue). A small superimposed signal from the mitochondria was also observed (data not shown). We confirmed the location of the Fe^2+^ signal, shown in Figure 3c, using correlative light and electron microscopy (CLEM).

### 2.2. WDR45 Mutation Alters the Lipid Metabolism, Mitochondrial Change and Autophagy in Fibroblasts

To investigate the molecular signatures affected by the *WDR45* mutation, we performed mRNA sequencing of patient derived *WDR45* mutant fibroblasts and normal fibroblasts. By comparing the mRNA expression profiles, we identified 2639 differentially expressed genes (DEGs), of which 1022 were upregulated and 1617 were downregulated between the two types of fibroblast cells (Figure 2A, Appendix A). To examine the cellular processes represented by the DEGs, we performed enrichment analysis of gene ontology biological processes (GOBPs). The downregulated genes were significantly (*p* < 0.05) associated with processes related to the cell cycle, lipid transport/metabolism, autophagy, vesicle-mediated transport and mitochondrial organization (Figure 2B). The upregulated genes were associated with processes related to apoptosis, response to oxygen-containing compounds, MAPK cascade, PI3K-Akt-mTOR signaling pathway and the Wnt signaling pathway (Figure 2C).

Consistent with our findings, previous studies have shown autophagic defects, diminished lysosomal function and mitochondrial abnormalities in *WDR45* mutant fibroblasts [7]. To understand the collective effects of the upregulated or downregulated processes, we constructed a network model describing the interactions between genes involved in these processes (Figure 2D). The network model showed downregulation of genes involved in autophagy (*TFEB*, *WIPI*, *ATG5*, *ATG12*, *SQSTM1*, *OPTN* and *BNIP3*), lysosomal membrane proteins (*VAMP4*, *RAB7B*, *MCOLN1*, *SLC17A5* and *LAPTM5*), and lysosomal acidic hydrolases (*CTSD*, *CTSL*, *CTSZ*, *CTSC*, *CTSK*, *CTSS*, *PLA2G15*, *NEU1* and *ACP5*), as well as genes involved in mitochondrial biogenesis and fatty acid metabolism (*PPARGC1A*, *APOE*, *SLC27A1*, *FABP3*, *ACSL4*, *ACSL5*, *CPT1C*, *ACAT1* and *ACADS*). This model also showed the upregulation of genes involved in mTOR-related signaling pathways (*AKT3*, *MAP2K3*, *MAPK11*, *STAT3* and *STAT4*).

Finally, we confirmed the differential expression of the representative genes involved in autophagy/lysosome (*TFEB*, *ACP5*, *CTSD*, *VAMP4*, *LIPA*, *LGMN*, *GAA* and *SORT1*), mitochondrial biogenesis/fatty acid oxidation (*PPARGC1A*, *SLC27A1*, *ACSL4*, *CPT1C*, *ACAT1* and *ACADS*) and mTOR-related signaling pathways (*AKT3*, *STAT3* and *STAT4*) in the *WDR45* mutant fibroblasts, compared with that in the normal fibroblasts (Figure 2E and Appendix A). Taken together, these data suggest that the loss of *WDR45* function increased iron accumulation caused by impaired autophagy flux and lysosomal and mitochondrial function in *WDR45* mutant fibroblasts.

### 2.3. Increase in Lysosomes Was Shown in WDR45 Mutant Human Fibroblasts and WDR45 Knockdown Cells

Puncta identified using lysotracker were bigger than those of normal fibroblasts under confocal microscopy (Figure 3a), and TEM also showed abnormally large vacuoles (Red arrows) in *WDR45* mutant cells (Figure 3b). There were 61.77 ± 41.15 (average number ± standard deviation) vacuoles in normal fibroblasts and 144.9 ± 48.16 in *WDR45* mutant fibroblasts (Figure 3b). A difference in the pH sensitivity can result in accumulation of GFP-LC3 in neutral autophagosomes and RFP-LC3 proteins in acidic autolysosomes. The number of autophagosome (YFP-LC3) puncta in each group was 38.76 ± 14.28 (normal fibroblasts) and 76.5 ± 26.69 (*WDR45* mutant fibroblasts). The number of autolysosome (RFP-LC3) puncta in each group was 20.3 ± 6.74 (normal fibroblasts) and 14.16 ± 6.48 (*WDR45* mutant fibroblasts). In *WDR45* mutant fibroblasts, the number of autolysosomes showing a red signal was decreased, suggesting dysfunction of the lysosomes. The levels of *ACP5* and *CTSD* in *WDR45* mutant fibroblasts were decreased (Figure 2E). To understand the way in which cellular iron handling might be involved, the precise location of iron accumulation was investigated using CLEM with lysotracker signals. CLEM showed the location to be abnormal autophagic vacuoles (Figure 3c). In comparison with lysosomes without iron accumulation, the ultrastructure of lysosomes with iron accumulation, including Fe^2+^ signal, showed less density.

### 2.4. Mitochondrial Fragmentation and Dysfunction and Increase in Lipid Droplets in the WDR45 Mutant Fibroblasts

We examined the function and morphology of mitochondria in *WDR45* mutant fibroblasts. The Mitotracker signal in each group was used to investigate the morphology of the mitochondria. Structural analysis revealed an increased number of fragmented mitochondria in *WDR45* mutant fibroblasts, in comparison with normal fibroblasts. (Figure 4a). The perimeter of the mitochondria was 3.2 ± 0.27 µm (normal) and 1.93 ± 0.68 µm (mutant fibroblasts). The area of the mitochondria was 0.36 ± 0.07 µm^2^ (normal) and 0.17 ± 0.02 µm^2^ (mutant fibroblasts). Image analysis was conducted using ImageJ [9]. Because the fluorescent signal was exaggerated, we also measured the perimeter of the mitochondria from EM images (the perimeter of the mitochondria in each was 0.83 ± 0.14 µm in control cells and 0.39 ± 0.04 µm in *WDR45* mutant cells). We analyzed the mitochondrial function in each group. The average basal respiration and ATP production were significantly decreased in *WDR45* mutants compared with normal cells. These data suggested that the presence of the *WDR45* mutation changed the mitochondrial bioenergetic function (Figure 4b). Using RNA analysis, we identified possible changes in the mitochondrial function associated with fatty acid oxidation. To investigate whether the *WDR45* deficiency contributed to mitochondrial impairment, including abnormal fatty acid oxidation, we observed lipid droplets using Lipi-Red probes under confocal microscopy in normal and *WDR45* mutant fibroblasts. A significant increase in the lipid signal in the *WDR45* mutant fibroblasts, compared with normal fibroblasts, was noted (Figure 4c). As shown in Figure 2B, lipid metabolism-related processes, including lipid transport and lipid storage, were downregulated in *WDR45* mutant fibroblasts. The abnormal accumulation of lipid droplets in the *WDR45* mutant fibroblasts could be related to their abnormal mitochondrial function.

## 3. Discussion

*WDR45* (also known as *WIPI4*) is one of the four mammalian homologs of the yeast gene *Atg18*, which has an important role in autophagy [10]. The WDR45 protein functions as a beta-propeller scaffold and appears to contribute to autophagy through its interactions with phospholipids and autophagy-related proteins. Several studies have reported that mutations in *WDR45* are associated with defects of early autophagy and ER stress. Reduced autophagic activity and the accumulation of aberrant early autophagic structures have been reported in a lymphoblastoid cell line from a patient with BPAN [2]. Specially, it was shown that the cells had abnormal regulation of autophagy, with an increase in the levels of LC3II [2]. Patient-derived fibroblasts have also been found to show autophagic defects [7]; furthermore, iron-containing macromolecules and organelles did not show degradation as a result of lysosomal defects [7]. However, the association between iron homeostasis and defects of autophagy related to the function of this protein was not fully understood.

In our previous study, loss of *WDR45* function due to the c.974-1G>A mutation, which was reported by our group [8], showed the induction of iron accumulation in the brains of a patient, and in the present study, we showed that iron accumulation occurred in the fibroblasts obtained from a patient (Figure 1). The functional role of *WDR45* in iron accumulation is mostly unknown, but iron accumulation by *WDR45* mutation induces oxidative stress and affects the cytosolic ROS, Ca^2+^, or cAMP levels [11,12]. The alterations of these cytosolic constituents change the activities of diverse signaling pathways and the expression levels of their target genes, thereby affecting cellular processes, such as autophagic flux, lipid metabolism, mitochondrial biogenesis, and apoptosis. Thus, understanding the alteration of gene expression programs caused by *WDR45* mutation will be important for understanding the pathogenesis of BPAN. Here, we used gene expression profiling to identify the molecular signatures and related organelles affected by the *WDR45* mutation in fibroblasts (Figure 2). The *WDR45* mutation resulted in changes in the mRNA levels of 2639 genes involved in various cellular processes. Among the DEGs, we identified 52 genes involved in autophagy-related cellular processes (Figure 2D and Appendix A). In our previous reports, the levels of P62, LC3 and TFEB were changed in the *WDR45* mutant, and Zao et al. also showed an increase in the levels of P62, LC31 and LC3II compared with those in the wild type in Nes-*Wdr45*fl/Y mice with impaired learning and memory [5]. There are other interesting genes that could be involved in autophagy or BPAN-related pathological features by *WDR45* mutation. For instance, the mRNA abundance of cathepsins (*CTSD, CTSL, CTSZ* and *CTSK*) was decreased by the *WDR45* mutation (Figure 2D,E). Autophagosome accumulation is reportedly caused by the reduced enzymatic activity of CTSD. The restoration of this protease in pathological fibroblasts results in recovery of the autophagic flux and lysosome homeostasis [13]. In addition, the mRNA abundance of several transcriptional factors (including *PPARGC1A* and *JUN*) was decreased by the presence of the *WDR45* mutation. In a recent study, abnormal autophagosomes were observed in *PPARGC1A* knockout mice; reduced P62 protein expression was also observed in *PPARGC1A* knockout mice, and was restored by *PPARGC1A* overexpression [14]. Thus, additional studies of the genes involved in autophagy and lysosomal function, the expression of which is altered by the *WDR45* mutation, may further reveal their functional roles in pathological fibroblasts.

To study the relationship between iron accumulation and cellular organelles, we analyzed the structural and functional changes in cellular organelles. The distinct changes included an increase in the number of lysosomal vesicles (Figure 3a), fragmented mitochondria (Figure 4a), and lipid droplets (Figure 4c) in *WDR45* mutant fibroblasts. Regarding autophagy, although there have been reports about iron and autophagy playing crucial roles in patients with the *WDR45* mutation, most discussed the early stages of autophagy. We found changes in mRNA expression related to all stages of autophagy. In particular, the activity of the late stages of autophagy was decreased in the fibroblasts obtained from a patient (Figure 2E). The quantification of the mRNA of genes related to lysosome activity showed a decrease in mutant fibroblasts (Figure 2E). Our RFP-GFP-LC3B observations in *WDR45* mutant fibroblasts also supported the contention that iron accumulation results in lysosomal dysfunction (Figure 3c). Observation of the precise location of iron accumulation showed that iron accumulated mainly in abnormal autophagic vacuoles; these observations also suggested the existence of a direct relationship between iron accumulation and impaired autophagic flux (Figure 1c and Figure 3). Jahng et al. also showed that iron overload inhibited late stage autophagic flux, with the accumulation of dysfunctional autolysosomes and loss of free lysosomes in skeletal muscle. Although they did not use the *WDR45* mutation, Jahng et al. suggested that iron overload represses mTORC1 activation and interferes with autophagic lysosomal regeneration [15]. Therefore, we investigated the difference between iron-laden and non-laden lysosomes. In our observation of fibroblasts from patients with iron overloads, the accumulation of enlarged vacuoles was observed; a similar result was observed within 24 h of iron treatment in the study by Jahng et al. [15]. Seiber et al. showed elevated levels of iron and reduced levels of L-ferritin in *WDR45* mutant fibroblasts, together with diminished lysosome function. Ferritin is an important protein for removal of excess iron from the cytoplasm, and helps store iron in a non-redox active form. Ferritin should be degraded in cells through the autophagic pathway. Xiong et al. reported that WDR45-deficient cells showed transferrin receptor accumulation, which affects autophagic degradation. They also suggested that *WDR45* mutation could promote ferroptosis by lipid peroxidation and ROS production [16]. ROS from iron metabolism can induce ferroptosis; it is also correlated with NADPH oxidase activity and lipid peroxidation products. Perturbations in MAPK can be major inducers of ferroptosis [17]. In our study, as with RNA sequencing, the MAPK-associated cascade was upregulated in *WDR45* mutant cells, and lipid metabolism process factors decreased (Figure 2B,C). However, we did not observe cell death in fibroblast cultures from the patient. We assumed that inhibition of ACSL4 (Figure 2E) could attenuate ferroptosis in the fibroblasts [18]. We also observed that cell growth did not change significantly (Appendix A) in the mutated fibroblasts.

Mitochondrial changes were another distinctive characteristic of *WDR45* mutant fibroblasts. Our results showed mitochondrial dysfunction and fragmentation (Figure 4). Iron participates in ROS generation and oxidative stress, which alters mitochondrial function. Zheng et al. showed an increase in ROS levels and mitochondrial dysfunction in mesenchymal stromal cells from patients with myelodysplastic syndrome [19]. Zheng et al. also showed that iron overload promotes mitochondrial fragmentation in the cells through the AMPK/MFF/Drp1 pathway. They also showed a reduction in ATP concentration, increase in ROS levels, and low electron transport chain II/III activity in the cells [19]. Our data show that ROS levels were increased in *WDR45* mutant fibroblasts, whereas they decreased after treatment with the iron chelator 2,2’-*Bipyridine* (Bpy) (Appendix A). In addition, mitochondrial dysfunction and fragmentation in *WDR45* mutant fibroblasts was significant compared with that in NHDF (Figure 4). Related genes were also downregulated, and the mRNA levels confirmed mitochondrial dysfunction (Figure 2). The results of stress-induced mitochondrial functional tests were correlated with previous results (Figure 4b).

Changes in lipid accumulation were found in the patient’s fibroblasts (Figure 4c). Maintenance of a functional population of mitochondria is ensured through effective lipid metabolism. Lipid metabolism is also involved in autophagic initiation under stress. Selective autophagy promotes lipid catabolism through degradation within the cell [20], possibly associated with iron accumulation. The iron overload-mediated disruption of lipid metabolism may promote mitochondrial dysfunction and autophagy dysregulation. A recent study reported that WDR45 deficiency had an important effect on the lipid transfer activity of *ATG2-WIPI* complexes in neurons [21]. In our study, *WDR45* mutant fibroblasts showed downregulation of genes related to lipid transport and lipid metabolism (Figure 2B,D). The results were in agreement with structural observations showing lipid accumulation in *WDR45* mutant fibroblasts.

In summary, our data showed that *WDR45* mutant fibroblasts from a patient with a c.974-1G>A mutation impaired autophagy and led to mitochondrial fragmentation and the accumulation of lipid droplets. These changes resulted in iron accumulation and may affect ferroptosis. Further studies will investigate whether these changes are regulated by the activation of mitochondria and lysosomes.

## 4. Materials and Methods

### 4.1. Preparation of Fibroblasts

A skin punch biopsy was obtained using a standard procedure [22]. Skin tissue obtained from the patient was washed three times with PBS, and then sliced to a size of 3–5 mm and transplanted into a collagen-coated culture dish. Tissue fragments were maintained in DMEM/10% FBS, with the medium changed every 2 days. After 7 days of explanting, tissue fragments were discarded and the sprouted fibroblasts were collected and used for experiments. Primary human adult dermal fibroblasts (NHDF) were purchased from LONZA (NHDF-Ad, CC-2511, CloneticsTM, Lonza, Basel, Switzerland), and used as control cells.

### 4.2. Cell Culture

Primary fibroblasts obtained from patients with the c.974-1G>A mutation and primary normal human dermal fibroblasts were cultured in DMEM (Gibco, Waltham, MA, USA) supplemented with 10% (*v*/*v*) FBS (Gibco, Waltham, MA, USA) and penicillin/streptomycin, and maintained at 37 °C in 5% CO_2_.

### 4.3. Measurement of WDR45 Expression Level

For Western blotting, cells were lysed with RIPA buffer (R0278, Sigma-Aldrich, St. Louis, MO, USA). Cell lysates were separated on 10% SDS-PAGE and transferred to a nitrocellulose membrane (Whatman, Maidstone, UK). Membranes were blocked with 5% non-fat milk in TBST for 30 min at room temperature. The membranes were incubated with primary antibodies overnight at 4 °C, and then with horseradish peroxidase-conjugated mouse- or rabbit-IgG for 1 h at room temperature. Antibodies were detected using an enhanced chemi-luminescence kit (Bio-Rad, Hercules, CA, USA). Primary antibodies against the following factors were used: Anti-WDR45 (19194-1-AP, Proteintech, Sankt Leon-Rot, Germany) and Anti-GAPDH (97166, Cell Signaling Technology, Danvers, MA, USA). Immunoblots were visualized and quantitatively analyzed using Fusion Fx Software (Vilber Lourmat, Marne-la-Vallee, France).

### 4.4. Confocal Microscopy Using Fe^2+^, Lipi-Red, Mitotracker, Lysotracker and GFP-LC3-RFP Plasmids

Cells were grown in 35 mm glass-bottomed culture dishes (801001, NEST, Wuxi, China) to 50–60% confluency. The next day, to visualize iron accumulation, cells were stained with 2 µM Fe^2+^ probe (SCT037, Merck Millipore, Billerica, MA, USA) for 15 min and then imaged under a confocal light microscope (Ti-RCP, Nikon, Tokyo, Japan). Fluorescence intensity was measured using ImageJ software version 1.51j8. Lipi-red (LD03, Dojindo, Kumamoto, Japan) 1 µmol/L, was used to measure the number of lipid droplets. To investigate the structural changes in the mitochondria, 200 nM of Mitotracker (M7514, Invitrogen, Carlsbad, CA, USA) signal was observed using confocal microscopy. Image analysis was carried out as detailed in Valente et al. [9] using ImageJ. To analyze structural changes in autophagy, cells were stained with 100 nM LysoTracker (L7525, Thermo Fisher Scientific, Waltham, MA, USA) for 15 min and then imaged under a confocal light microscope. The puncta identified using lysotracker were compared in each group. To analyze lysosome function, autophagy tandem sensor RFP-GFP-LC3B Kits (P36239, Thermo Fisher Scientific, Waltham, MA, USA) were used, and the numbers of puncta of the green and red signal in each group were compared. All images were taken in live cells without any fixation.

### 4.5. mRNA Sequencing and Data Analysis

Total RNA was obtained from patient-derived *WDR45* mutant fibroblasts and normal fibroblasts. The integrity of the total RNA was analyzed using a 2100 Bioanalyzer (Agilent Technologies, Santa Clara, CA, USA). The RNA integrity number values for all the samples were larger than 9.3. Poly(A) mRNA isolation from the total RNA and subsequent fragmentation were performed using TruSeq RNA Sample Prep Kits v2 (Illumina, San Diego, CA, USA), according to the manufacturer’s instructions. The adaptor-ligated libraries were sequenced using an Illumina NovaSeq 6000 (Macrogen Inc., Seoul, South Korea). mRNA sequencing was performed for three biological replicates of each condition. From the resulting read sequences for each sample, adapter sequences (TruSeq universal and indexed adapters) were removed using the Cutadapt software (version 2.7; [23]). The remaining reads were then aligned to the *Homo sapiens* reference genome (GRCh38) using TopHat2 software (version 2.1.1) (https://ccb.jhu.edu/software/tophat/index.shtml, accessed on 1 June 2021) with default parameters (Appendix A) [24]. After the alignment, we counted the numbers of reads mapped to the gene features (GTF file of GRCh38.89) using HTSeq [25]. Read counts for the samples in each condition were then normalized using trimmed mean of M-values normalization of the edgeR package [26]. The raw data were deposited into the Gene Expression Omnibus (GEO) database with the accession ID: GSE184520.

### 4.6. Identification of Differentially Expressed Genes

The number of reads for the gene features were converted to log_2_-values after adding one (pseudo count) to the read counts. To identify DEGs, hypothesis testing was performed [27]. Briefly, for each gene, a *T*-statistic value was calculated using Student’s *t*-test for the comparison of *WDR45* mutant and normal fibroblasts. An empirical distribution of the *T*-statistic value for the null hypothesis (i.e., the genes are not differentially expressed) was then estimated by performing all possible combinations of random permutations of samples. The *p*-values from the Student’s *t*-test for each gene were computed using a two-tailed test with the empirical null distribution. False discovery rates (FDRs) for the *p*-values were then computed using the Storey method [28]. The DEGs were identified as the genes that had FDRs <0.05 and fold changes >1.5. To identify the cellular processes represented by the DEGs, enrichment analysis of GOBPs was performed using DAVID software [29] and the GOBPs with a *p*-value <0.1 were selected as the processes represented by the DEGs. The network model was reconstructed for the selected DEGs using Cytoscape software [30]. The nodes in the network model were arranged based on the locations and relationships of the corresponding genes in the Kyoto Encyclopedia of Genes and Genomes (KEGG) pathway database [31].

### 4.7. mRNA Quantification

For quantitative real-time reverse-transcription PCR (PT-qPCR), RNA was harvested using TRIzol lysis buffer (Invitrogen, Carlsbad, CA, USA), followed by DNase I treatment, and first-strand cDNA synthesis was performed using 1 ug of total RNA and oligo (dT) for reverse priming with SuperScript III Supermix (Invitrogen, Carlsbad, CA, USA). Amplification real-time PCR was performed using a SYBR Green PCR Master Mix (Applied Biosystems, Foster City, CA, USA). Each PCR reaction contained cDNA at a 10-fold dilution, and gene-specific primers. The thermal cycle used was 2 min at 50 °C, 10 min at 95 °C, and 40 cycles of 15 s denaturation at 95 °C with 1 min annealing at 60 °C. The mean cycle threshold (CT) values were calculated, with normalization to GAPDH as an internal control. Samples were analyzed using quantitative real-time PCR with gene-specific primer pairs, on an ABI 7500 fast real-time PCR detection system (Life Technologies, Foster City, CA, USA) using the ΔΔCT method [32]. In each case, multiple reactions were performed using 2–3 independent biological replicates, with the primers listed in Appendix A.

### 4.8. Localization of Iron Accumulation Using Correlative Light and Electron Microscopy (CLEM)

CLEM was performed as previously described [33]. The cells were grown in 35 mm glass grid-bottomed culture dishes (801001, NEST Biotechnology Co., Wuxi, China) to 50–60% confluency. Cells with Fe^2+^ probe treatment were imaged using confocal microscopy, and then fixed with 2.5% glutaraldehyde (16210, Electron Microscopy Sciences, Hatfield, PA, USA) and 2% paraformaldehyde (19210, Electron Microscopy Sciences, Hatfield, PA, USA) in 0.1 M cacodylate solution (pH 7.0; C0250, Merck). After being washed, the cells were dehydrated with a graded ethanol series and infiltrated with an embedding medium. After embedding, 60 nm sections were cut horizontally to the plane of the block (UC7; Leica Microsystems, Wetzlar, Germany) and were mounted on copper slot grids with a specimen support film. Sections were stained with uranyl acetate and lead citrate. The cells were observed at 120 kV in a Tecnai G2 microscope (Thermo Fisher Scientific, Waltham, MA, USA). Confocal micrographs were produced as high-quality images using PhotoZoom Pro 8 software (Benvista Ltd., Houston, TX, USA). Enlarged fluorescence images were fitted to the electron micrographs using the ImageJ BigWarp program.

### 4.9. Electron Microscopy Analysis

Ultrastructural analysis was conducted using transmission electron microscopy (TEM). Cells in each group were grown on coverslips. The samples were immediately fixed with 2.5% glutaraldehyde-mixed 2% paraformaldehyde solution for 1 h, followed by post-fixation in 2% osmium tetroxide (OsO_4_; 19208, Electron Microscopy Sciences, Hatfield, PA, USA) for 1 h at 4 °C. The block was stained in 2% uranyl acetate (15200, Electron Microscopy Sciences, Hatfield, PA, USA) and dehydrated with a graded ethanol series. The samples were then embedded into epoxy medium (EMS, Hatfield, PA, USA). Embedded samples were sectioned (60 nm) with an ultra-microtome (UC7, Leica Microsystems, Wetzlar, Germany), and the sections were then viewed on a Tecnai 20 TEM (Thermo Fisher Scientific, Waltham, MA, USA) at 120 kV. They were then double stained with UranyLess (22409, Electron Microscopy Sciences, Hatfield, PA, USA) for 2 min and 3% lead citrate (22410, Electron Microscopy Sciences, Hatfield, PA, USA) for 1 min. Images were captured with a US1000X-P camera 200. The acquired images were stitched together using Photomontage software (Thermo Fisher Scientific, Waltham, MA, USA). The number of abnormal autophagic vacuoles of larger size was counted manually.

### 4.10. Measurement of Mitochondrial Bioenergetics Using a Seahorse XF24 Analyzer

Using a Seahorse XF Extracellular Flux Analyzer XFe24 (Seahorse Bioscience, Lexington, MA, USA), we could measure mitochondrial bioenergetics in real time in our fibroblast cell lines. Briefly, normal fibroblast cells and *WDR45* mutant cells were seeded at 50,000 cells per well into Seahorse Bioscience XFe24 cell culture microplates. Prior to the assay, XFe24 sensor cartridges were hydrated in a non-CO_2_ incubator overnight. On the next day, cells were equilibrated in XF assay medium containing 4 mM glutamine and 1 mM sodium pyruvate, supplemented with 10 mM glucose in a non-CO_2_ incubator for 1 h. The mitochondrial processes were interrogated by sequential injection of oligomycin (1 µM), carbonyl cyanide 4-(trifluoromethoxy) phenylhydrazone (FCCP, 2 µM) and rotenone/antimycin A (0.5 µM). Measurements were performed in four independent experiments. All data were expressed as mean ± SEM (standard error of the mean). Statistical analyses were performed using GraphPad Prism^®^ software v. 7.00 (GraphPad Software, San Diego, CA, USA, www.graphpad.com). Comparisons between two values were performed using unpaired Student’s *t*-tests.

### 4.11. Statistical Analysis

Densitometric analysis was performed using ImageJ. Statistical analysis was performed using GraphPad Prism software. All data were normalized by the mean values of the replicate data in normal fibroblasts (control). Data are shown as mean ± standard deviation or SEM, unless otherwise indicated. Unpaired Student’s *t-tests* were used to compare means between groups. A *p*-value <0.05 was considered statistically significant. (** *p* < 0.01, *** *p* < 0.001 and **** *p* < 0.0001).

## Figures and Tables

**Figure 1 ijms-22-11650-f001:**
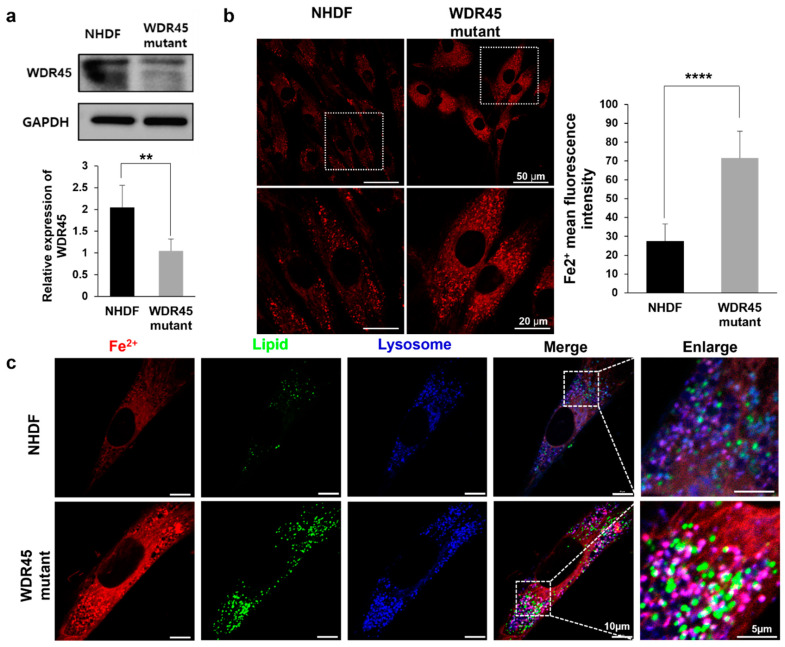
Increase in iron overload in *WDR45* mutant fibroblasts. (**a**) Abundance of *WDR45* in mutant and normal fibroblasts, with GAPDH as loading control for Western blotting (*n* = 4). (**b**) Fluorescence of Fe^2+^ (2 µM, 30 min) showing puncta of iron accumulation (*n* = 50). The scale bars represent 50 µm (top) and 20 µm (bottom) in the magnified images. (**c**) Imaging under confocal microscopy using Fe^2+^, Lipi-Red and lysotracker. The puncta of lipid (green color) and lysotracker (blue color) in each *WDR45* mutated cell and normal fibroblast were compared with the location of the puncta identified by the Fe^2+^ probe. Many puncta identified using lysotracker were superimposed on the Fe^2+^ signal (red color). The superimposed signals from Fe^2+^ and lysotracker are shown in pink. The scale bars represent 10 µm and 5 µm in the magnified images. ** *p* < 0.01, **** *p* < 0.0001 by Student’s *t*-test.

**Figure 2 ijms-22-11650-f002:**
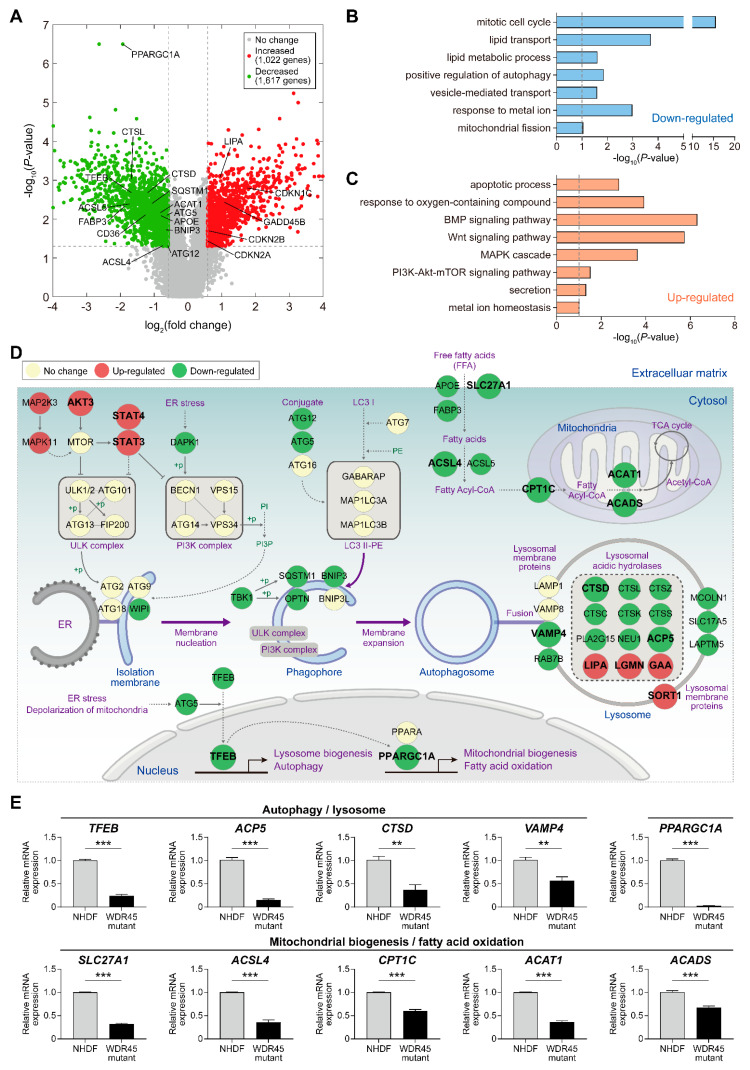
Cellular processes affected by *WDR45* mutation in fibroblasts. (**A**) Volcano plot showing differentially expressed genes (DEGs) in *WDR45* mutant fibroblasts compared with the normal fibroblasts. The X- and Y-axes represent log_2_-fold-change and –log_10_ (*p*-value), respectively. Red and green dots represent upregulated and downregulated genes, respectively. Gray dots represent genes whose expression was not significantly different. (**B**,**C**) Gene ontology biological processes (GOBPs) represented by the downregulated (**B**) and upregulated (**C**) genes. The dotted line indicates the *p*-value cutoff used. (**D**) Network model describing the interactions among cellular processes represented by the DEGs. The node colors represent upregulation (red), downregulation (green) and no change (yellow) of the corresponding genes in *WDR45* mutant fibroblasts compared with normal fibroblasts. Nodes are arranged and connected according to the activation (arrows) and inhibition (suppression symbols) information in the KEGG pathway database. “+ *p*,” phosphorylation. (**E**) Relative mRNA levels of the indicated representative genes in the network model were analyzed by quantitative RT-PCR. mRNA level of each gene was normalized using the GAPDH level. Data obtained from more than two independent experiments (*n* = 2–4) are presented as the mean ± SEM. ** *p* < 0.01; *** *p* < 0.001 by Student’s *t*-test.

**Figure 3 ijms-22-11650-f003:**
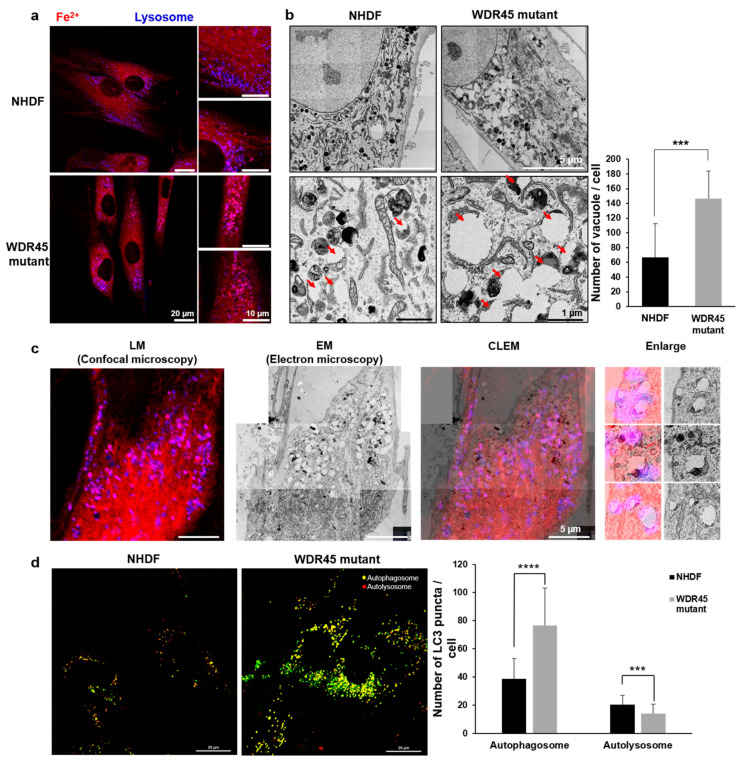
Abnormal autophagy was increased in *WDR45* mutant fibroblasts. (**a**) Increase in the lysotracker signal in *WDR45* mutant fibroblasts. The lysotracker signal (blue) is superimposed on the Fe^2+^ probe signal (red). *WDR45* mutant fibroblasts showed increased pink (superimposed) signal. (**b**) TEM images showing increased abnormal autophagic vacuoles in *WDR45* mutant fibroblasts. Red arrows represent abnormally large autophagic vesicles. Size bar = 5 μm (top panels) and 1 μm (bottom panels). *** *p* < 0.001 (*n* = 10). (**c**) The location of the pink signal under confocal microscopy was specifically in the abnormal autophagic vacuoles under electron microscopy. TEM images superimposed on confocal images show iron accumulation in the abnormal autophagic vacuoles. (**d**) Fibroblasts infected with baculovirus. RFP-GFP-LC3 (10 MOI) showed changes in lysosomal activity under confocal microscopy. Representative overlay images are shown. Yellow puncta indicate autophagosomes, and red puncta represent autolysosome-dependent pH differences. Size bar = 20 µm. *** *p* < 0.001, **** *p* < 0.0001 (*n* = 30). The number of puncta identified using GFP/RFP-LC3 per cell was analyzed using ImageJ.

**Figure 4 ijms-22-11650-f004:**
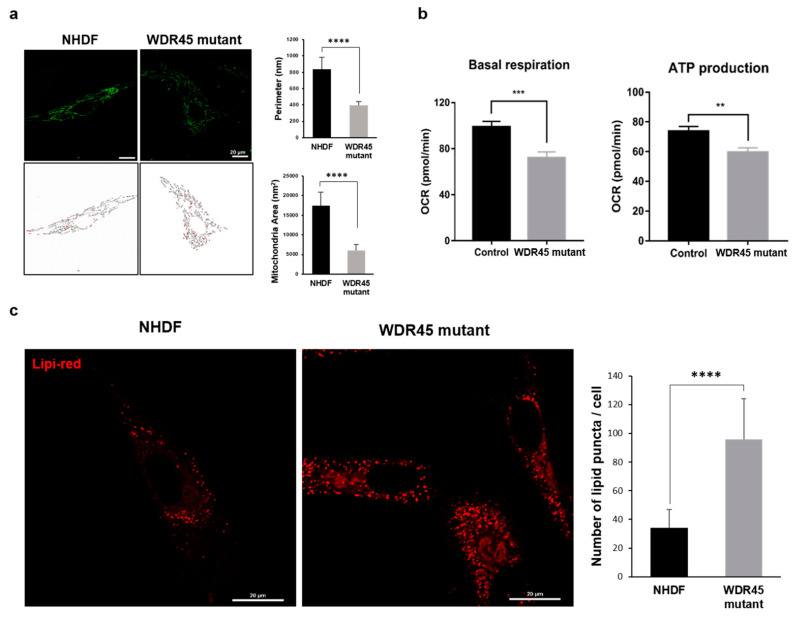
Dysfunction of mitochondria was increased in *WDR45* mutant fibroblasts. (**a**) Confocal imaging of cells treated with 200 nM Mitotracker for 30 min showed mitochondrial fragmentation. The scale bar is 20 µm (*n* = 50). The area and perimeter of the mitochondria were decreased in the *WDR45* mutant fibroblasts. (**b**) Basal respiration and ATP production in normal and *WDR45* mutant cells were calculated using Seahorse profiling. Oligomycin: 1 µM, FCCP: 2 µM, Rotenone + antimycin A: 1 µM. (**c**) Lipi-Red signal was increased in *WDR45* mutant fibroblasts. The number of puncta of lipid droplets (red) was increased in the *WDR45* mutant cells compared with normal fibroblast cells. Scale bars represent 20 µm (*n* = 50). ** *p* < 0.01, *** *p* < 0.001, **** *p* < 0.0001 by Student’s *t*-test.

## Data Availability

The data sets generated during the current study are available from the corresponding authors on reasonable requests.

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
