# Peer review of "Iron Accumulation and Changes in Cellular Organelles in WDR45 Mutant Fibroblasts"

_ijms, 2021, doi:10.3390/ijms222111650_

Round 1

Reviewer 1 Report

The manuscript entitled “Iron accumulation and changes of cellular organelles in WDR45 mutant fibroblasts” by Lee HE et al. to IJMS describes the confirmation of previously reported Iron accumulation phenotype and extended study further by using mRNA sequencing approach along with various microscopic techniques to report abnormalities in cellular organelles including mitochondrial defects, lipid metabolism and autophagy fluxes. The study conclusions are supported partially by the results and adds a very little knowledge to the community with leaving lot of mechanistic questions unanswered. The reviewer feels that the findings are less for an ‘Article’ and can be submitted as ‘Communications’ type article. The reviewer enthusiasm remains limited due to the following concerns:

  • Introduction: Though 977-1G>A mutation was reported earlier, authors should consider to provide some important details such as exon location and functional consequence of this SNV (such as splicing variant-shorter or larger protein..?) so that the readers get the content reported in the paper.
  • Other than Organelles dysfunction, It is unclear about the cellular phenotype due to iron overload in mutant cells such as growth, cell cycle, ROS, apoptosis etc. The authors can report how the mutant cells behave in vitro prior to describing intra-cellular phenotypes.
  • There is no attempt from authors to address what signaling pathways driven the organelles phenotype in WDR45 mutant cells. There are several signaling pathways upregulated in mRNA seq data (Figure 2B) and can be pursued for further validation. Though it might be out of lab scope, it will be interesting to see the mechanisms behind the phenotype the authors reporting in this paper.
  • As authors indicated, Figure 1 was just to confirm the phenotype of cells published earlier (do not provide anything new) and can be moved to supplementary instead.
  • Authors indicated ‘Significant’ decrease of WDR45 levels (p.2; line85/86), but the error bars (n=?) and statistics were missing in Figure 1A.
  • Authors concluded that these defects in different organelles were due to Iron overload. However, there are no findings or experiments to support that hypothesis. Authors could test few Iron scavengers treatment (to decrease iron over load) or some antioxidants (ROS scavengers, GSH, ascorbic acid.. etc) to overcome the ROS due to iron overload and reversal of the phenotypes in organelles to link it with iron metabolism in mutant cells. Vice versa, use chemical agents to increase iron overload in control cells to match phenotype with that of mutant cells as positive control.
  • Does authors see any upregulation of Fe transporters in mutant cells? Also, please add few sentences about the role of WDR45 in gene/transcriptional regulation if any in Introduction or Discussion.
  • Please indicate whether the control cells (NHDF) used are matching the control to a case.
  • In case of mRNA sequencing, Does authors submitted the Seq data files to NCBI/GEO? Please provide the accession number in the methods or share the private data link if any for review purpose. Also, provide the details such as number of reads acquired for each from replicate/avg reads per group.
  • The NCBI Clinivar recommends the use of ‘WDR45 c.974-1G>A’ instead of 977-1G>A as authors writing. Authors can mention this and better to use preferred name for clarity.
  • Before the current authors, this mutation was reported first by Liu et al. (2018) in Chin.Med.J. 131 (24) 2991 in infant diagnosed with BPAN. So, please reconsider to cite this article and rewrite sentence ‘we reported… new mutation.. in p.2 line.71-72).
  • The n=10 is very less for a microscopic quantifications as in Figure 1B. Also, similar with Figure 3B (n=5) and #D (n=6). Please consider adding data from some more cells to support the phenotype.
  • Please consider qPCR validation of few upregulated genes in mRNA seq as well (Figure 2E).
  • In Figure 4A, the bottom left NHDF image do not correlate to the mitotracker image shown above. WDR45 mutant images overlay each other. Check for accuracy.
  • Please provide n=? in figure 4B & 4C legend and add more cells (50-100) in 4C to see biological variance.
  • Correct ‘GRCm38’ in p.12; line 371 to ‘GRCh38’.
  • Indicate the method used for computing adjusted p-values in p.12; line 384/385. Is it Bonferroni- or other test?
  • Please cite the paper (PMID: 11846609) for ddCT method (p.13; line407).
  • There are several statements in the manuscript needs to be cited with proper references. Please carefully read over and cite the appropriate.

Reviewer 2 Report

In the abstract, the authors state that: Iron overload in the brain, defined as excess stores of iron, is known to be related to neurological disorder. Among neurodegeneration with brain iron accumulation, we reported a specific point mutation, 977-1G>A in WDR45, showing iron accumulation in the brain, and autophagy defects in the fibroblasts. The introduction also shows evidences that iron plays important roles in brain homeostasis and that mutations in the autophagy gene WDR45 are emerging cause of neurodegeneration with brain iron accumulation. But the authors did their work in fibroblasts that carry the 977-1G>A point mutation in the gene WDR45.

I would like to know why these experiments were done on fibroblasts instead of brain cells.

Where is the 977-1G> A mutation located in the WDR45 gene? How does the mutation change gene expression?

Line 93-94 “In figure 1C, major Fe2+ signal (red) was superimposed with lysotracker (blue), not lipid signal (green).” It would be more correct to state that the location of the Fe2 + signals are compatible with those of the lysotracker.

Line 111: To investigate the molecular signatures affected by the WDR45 mutation, we performed mRNA sequencing of patient-derived WDR45 mutant fibroblasts and normal fibroblasts. The methodology for obtaining fibroblasts from patients is not described. In addition, the authorization of the ethics committee is not presented.

 Line 163. A difference in the pH sensitivity can cause varying degrees of accumulation of GFP-LC3 and RFP-LC3 proteins in neutral autophagosomes and acidic autolysosomes. Autophagosome formation is a dynamic process and I have some doubts that LC3 when fused with GFP accumulates in neutral autophagosomes and LC3-RFP only in acidic autolysosomes.

Line 194-195 Mitochondrial network analysis revealed an increased number of fragmented mitochondria in WDR45 mutant fibroblasts, compared to normal fibroblasts.  (Figure 4a). The perimeter of the mitochondria was 3.2 ± 0.27 μm (normal) and 1.93 ± 0.68 196 μm (mutant fibroblasts). In my experience, confocal microscopy is not sensitive enough to allow differences in the size of mitochondria.

CLEM and TEM methodology are better described than the confocal microscopy method. Were the cells fixed for confocal microscopy?

Reviewer 3 Report

In their paper entitled “Iron accumulation and changes of cellular organelles in WDR45 mutant fibroblasts”, the Authors report that mutations in the WD Repeat Domain 45 (WDR45) gene causes iron accumulation in fibroblasts, with a series of effects on different cellular organelles, and on the process of autophagy. As a consequence of these alterations, significant changes in metabolism were also found.

The article is clearly written and of interest. However, some points deserve attention before acceptance:

  1. First of all, it is not clear whether the present study relies on the mutant fibroblasts coming from only one patient or whether more patients with the same mutation have been enrolled. See, for example, lines 68-70 (in Introduction): “Therefore, we investigated iron accumulation in WDR45 mutant fibroblast cells from patients, and studied the relationship between iron accumulation, defects of autophagy, and mitochondrial dysfunction through structural and functional analysis”, as well as lines 236-239 (Discussion): “In our previous study, loss of WDR45 function due to the c977-1G>A mutation, which was newly reported by our group [7] , showed the induction of iron accumulation in patients’ brains, and in the current study we have shown iron accumulation in fibroblasts from patients”. On the other hand, see: lines 72-75 (Introduction): “In this study, we found major iron accumulation in specific lysosomes of WDR45 mutant fibroblasts from the patient, and a little of accumulation of iron was observed in  mitochondria with the 977-1G>A mutation”, as well as lines 88-80 (Results): “we measured the puncta of a fluorescent Fe2+ probe for the analysis of iron accumulation in WDR45 mutant fibroblasts from the patient using confocal microscopy”, and lines 293-294 (Discussion): “However, we did not observe cell death in fibroblast culture from the patient”. Although these discrepancies could be due to spelling mistakes, and although clear differences have been found in mutant fibroblasts in comparison with fibroblasts from healthy people, this point should be clarified by stating the number of patients. Then, if the patient was only one, this limitation should be clearly acknowledged and discussed;
  2. Please report the complete name of the protein WDR45 the first time it is cited;
  3. Abstract: the sentence “We first identified the main location of iron accumulation in the mutant fibroblasts and then investigated the effects of this accumulation on other organelles, including lipid droplets, mitochondria, and lysosomes” (lines 18-20) suggests that the main location of iron accumulation is not one of the organelles cited in it; however, immediately below (lines 23-24), it is reported that “The majority of iron accumulation occurred in the lysosomal vesicles”;
  4. Results, line 88: “Iron accumulation in the patient’s brain was confirmed [7]”: this sentence refers to clinical data available for the patient, and not to the present study: this point should be briefly explained.

Round 2

Reviewer 1 Report

Dear authors,

The revised manuscript with added data and facts has improved significantly and I appreciate the efforts made by authors to address most of reviewer concerns. 

Minor comments: 

1) Authors stated in one of the responses that they cited Liu et al. (2018) in Chin.Med.J. 131 (24) 2991. But, reviewer failed to found this citation in the revised version. Please add this citattion to give credit for their work to report the mutation first by Liu et al. (2018) in Chin.Med.J. 131 (24) 2991 in infant diagnosed with BPAN. So, please reconsider to cite this article and rewrite sentences in Introduction. 

Best wishes,

Author Response

Thank you very much for your comment.

We will double-check to make sure the citation. 

Reviewer 2 Report

Authors have answered to all my questions and now It can be accepted for publication

Author Response

Thank you for your positive review.